# Impact of Ethyl Methane Sulphonate Mutagenesis in *Artemisia vulgaris* L. under NaCl Stress

**DOI:** 10.3390/biotech10030018

**Published:** 2021-08-21

**Authors:** Sudheeran Pradeep Kumar, B.D. Ranjitha Kumari

**Affiliations:** 1Department of Postharvest Science of Fresh Produce, Agricultural Research Organization (ARO), The Volcani Center, P.O. Box 15159, HaMaccabim Road 68, Rishon Lezion 7505101, Israel; 2Department of Botany, Bharathidasan University, Tiruchirappalli 620024, India; Ranjithakumari2004@yahoo.com

**Keywords:** antioxidants, *Artemisia vulgaris* L., ethyl methane sulphonate (EMS), mutagenesis, salt-tolerance, superoxide dismutase (SOD)

## Abstract

The present investigation aimed to obtain salt-tolerant *Artemisia vulgaris* L. to develop a constant form through in vitro mutagenesis with ethyl methane sulphonate (EMS) as the chemical mutagen. NaCl tolerance was evaluated by the ability of the callus to maintain its growth under different concentrations, ranges from (0 mM to 500 mM). However, NaCl salinity concentration at (500 mM) did not show any development of callus, slight shrinking, and brown discoloration taking place over a week. Thus, all the biochemical and antioxidant assays were limited to (0–400 mM) NaCl. On the other hand, selected calluses were treated with 0.5% EMS for 30, 60, and 90 min and further subcultured on basal media fortified with different concentrations of 0–400 mM NaCl separately. Thus, the callus was treated for 60 min and was found to induce the mutation on the callus. The maximum salt-tolerant callus from 400 mM NaCl was regenerated in MS medium fortified with suitable hormones. Biochemical parameters such as chlorophyll, carotenoids, starch, amino acids, and phenol contents decreased under NaCl stress, whereas sugar and proline increased. Peroxidase (POD) and superoxide dismutase (SOD) activities peaked at 200 mM NaCl, whereas catalase (CAT) was maximum at 100 mM NaCl. Enhanced tolerance of 0.5% the EMS-treated callus, attributed to the increased biochemical and antioxidant activity over the control and NaCl stress. As a result, the mutants were more tolerant of salinity than the control plants.

## 1. Introduction

*A. vulgaris* L. (mugwort) is a medicinally useful traditional plant and is widely being used for the healing of diabetes. Its extract is used for epilepsy and various combinations for psychoneurosis, irritability, depression, sleeplessness, and anxiety stress, among other things [1]. The plant is also a highly efficient antidote to insect or pest poison. In addition, some of the antibacterial potentials and phytochemical screenings were evaluated for the medicinal plant such as *Vitex negundo* L. against plant pathogens that were previously reported [2]. The in vitro regeneration of the high artemisinin-producing somaclonal variation against (NaCl) salt-tolerance and the improvement of sequence-characterized amplified region (SCAR) markers in ‘*Artemisia annua* L.’ were reported by [3].

Salt stress is a major environmental limitation that has a global impact on plant yield and distribution. Salinity (NaCl) inhibits plant growth by altering homeostasis in water status and ionic distribution and causing oxidative stress. Due to a rise in salt concentration, around 20% of the world’s framed surfaces are considered non-fertile [4]. The semiarid and arid zones are especially prone to environmental concerns [5]. Ion toxicity, osmotic pressure, mineral shortages, and a mix of physiological and biochemical abnormalities are all common symptoms of salt stress [6]. The plant cell adaptation and variation towards high level salinity involve osmotic change and the compartmentalization of toxic ions, whereas an increasing body of evidence suggests that high salinity also stimulates the formation of oxidative stress and reactive oxygen species (ROS) [7]. 

Plants have developed several defense mechanisms, and one of them is an antioxidant system and capability of surviving harmful oxidation in situations that favour the formation of oxidative stress, such as salt stress. Plant cells have developed a sophisticated antioxidant system that comprises both low molecular-mass antioxidants andreactive oxygen species (ROS)-scavenging enzymes, together with peroxidase (POD), superoxide dismutase (SOD), and catalase (CAT) [8]. Furthermore, plants continuously produce reactive oxygen species (ROS) as a by-product of many physiological, metabolic pathways, such as photo-respiration, photosynthesis, and CO_2_ absorption. Whether ROS behave as harmful, protective, or signaling components is determined by the delicate balance between ROS production and scavenging mechanisms to adjust the correct spot and timing [9]. The most vital enzymatic antioxidant is superoxide dismutase (EC 1.15.1.1); it catalyzes the dismutation of the O_2_ into H_2_O_2_, which is then deactivated to water by peroxidase (EC 1.11.1.7) and catalase (EC 1.11.1.6).

In crop development, mutation initiation has emerged as a novel strategy in crop improvement to supplement the technical limitations of conventional breeding in improving cultivars for specific attributes. According to previous accounts, more than 2500 enhanced mutant varieties have been released for marketable production [10]. Mutagenesis has become popular in recent decades due to its ease of use, low cost, applicability to all plant species, and ability to be used on a small or large scale. The frequency of produced mutations can be matched by adjusting the mutagen dose, and saturation can be easily reached [11]. In vitro culture is a good technique for the selection of salt-tolerant mutants, as it is carried out under controlled conditions with limited time [12]. Furthermore, unlike the entire plant, many numerous lines can be screened for the required feature at the same time. Salt-tolerant cell lines and plants have been derived from plant tissue culture techniques in a variety of species, including rice, wheat [13,14].

Plants that are tolerant to biotic or abiotic stress have a variety of resistance mechanisms, such as osmoregulation, ion homeostasis, and antioxidant and hormonal systems, serving plants to stay alive and develop prior to their reproductive stages [15]. To accomplish salt tolerance, plant cells evolve numerous biochemical and physiological traits to achieve salt tolerance. The in vitro tissue culture system is effective for assessing tolerance to environmental stresses due to the stress conditions. In vitro tissue culture is a useful technique for studying the physiological and biochemical systems that function at the cellular level in response to stress [16,17].

Cell line collection and subsequent plant regeneration have been employed extensively in the production of salt-tolerant plants [18,19]. These findings were new for the ethyl methane sulphonate (EMS) or other chemical mutagens used on callus and whole plant regeneration of *A. vulgaris* for salt-tolerance. The goal of this work is to use the EMS in vitro approach and plant regeneration to create a salt-tolerant mutant.

## 2. Materials and Methods

### 2.1. Seed Germination and Explant Establishment

The seeds of *A. vulgaris* were obtained from the company Johnny’s Selected Seeds, located at Winslow, Maine, USA. Seeds were surface sterilized and germinated on MS media containing 10% filter sterilized coconut water. A total of 95 percent of the seeds germinated after 5–7 days of dark exposure at 23 °C. Fully developed seedlings were observed within 30 days of being switched to photoperiodic settings conditions (16/8 h light/dark) at 25 ± 1 °C, performed accordingly by [20]. The explants for this investigation were taken from nodal segment explants were excised from 35-day-old in vitro grown seedlings.

### 2.2. Callus Initiation

Explants were carefully dissected (nodal) from 35-day-old in vitro seedlings and cultured in basal nutrient medium Murashige and Skoog (MS), supplemented with B_5_ vitamins 3percent (*w*/*v*) sucrose, 4.52 µM 2,4-Dichlorophenoxy acetic acid (2–4, D), and 2.65 µM α-Naphthalene acetic acid (NAA) (Himedia, Mumbai, India), for callus induction [21,22]. The pH of the medium was adjusted to 5.7 (supplemented with a growth regulator) with 1 N NaOH or 1 N HCl (Himedia, Mumbai, India), before gelling with 0.8percent agar (Himedia, Mumbai, India), and autoclaved at 121 °C for 15 min at 15 lbs pressure. The explants were placed on the culture medium horizontally. All the cultures were incubated at 25 ± 1 °C for optimum growth and development. Two subcultures were performed at 10-day time intervals.

### 2.3. Assessment of NaCl Tolerant

After 25 days, the viable callus was chopped into small pieces of 5–6 mm size and further inoculated on a callus initiation medium supplemented with varied NaCl concentrations (0, 100, 200, 300, 400 and 500 mM). Subcultures of these culture mediums were performed three times at 10-day intervals. The survival rate of callus growth was assessed in all NaCl concentrations ranging from (0 to 500 mM). High salinity at 500 mM NaCl was found to be fatal for the callus development and at 400 mM NaCl, the callus can survive for more than 30 days. Therefore, 400 mM NaCl was chosen as the proper concentration for this present investigation.

### 2.4. Ethyl Methane Sulphonate (EMS) Treatment

After 25 days of callus induction, it was cut into small pieces, weighed, and treated with 0.5% EMS solution (made in sterilized distilled water and membrane filtered) for 30 min, 60 min, and 90 min, respectively, and inoculated on MS Basal media containing varied concentrations of NaCl (0 to 400 mM), B_5_ vitamins, 3percent (*w*/*v*) sucrose, 5.42 µM of 2 (2–4, D) and 2.65 µM of (NAA). Three subcultures were carried out at 10-day intervals between them. Finally, calluses were collected for further regeneration with a 400 mM NaCl concentration.

### 2.5. Shoot Induction and Elongation

For shoot induction and regeneration, EMS-treated calluses were transferred to a basal medium containing B_5_ vitamins, supplemented with 3percent (*w*/*v*) sucrose, 4.44 µM of N^6^-benzyladenine (BA), and 2.78 µM of Thidiazuron (TDZ) (Sigma-Aldrich, Bangalore, India) fortified with the selection pressure of 400 mM NaCl [21]. After that, multiple shoots were then transferred to the shoot elongation medium containing MS basal salts, B_5_ vitamins, 3percent (*w*/*v*) sucrose, and 1.44 µM of Gibberellic acid (GA_3_) (Sigma-Aldrich, Bangalore, India).

### 2.6. Rooting and Acclimatization of Plantlets

After 2–3 weeks, elongated shoots were removed from the elongation medium and transferred to MS media supplemented with various concentrations of Indole-3-acetic acid (IAA), (Himedia, Mumbai, India), (2.85, 5.70, 8.55, and 11.40 µM). Plantlets with well-developed roots were removed from the rooting media and rinsed the adventitious roots-gently under running tap water to remove the adhering material; plantlets were then transferred to artificial cups (10 cm diameter) containing with autoclaved garden soil, farmyard soil, and sand (2:1:1) for hardening. Plantlets were kept in a culture environment maintained at (25 ± 1 °C) conditions. Each plantlet was sprayed with distilled water every two days for 3 weeks, followed by tap water for 2 weeks and transferred to conventional laboratory conditions.

### 2.7. Biochemical Analysis

#### 2.7.1. Evaluation of Chlorophyll and Carotenoid

Chlorophylls (a + b) and carotenoid contents were determined by extracting pigments in 80% acetone (Himedia, Mumbai, India), and calculated as mg/g FW as described by [18].

#### 2.7.2. Evaluation of Total Soluble & Reducing Sugars, and Starch

The anthrone–sulphuric acid method was used to quantify total soluble sugars and starch using 0.2% anthrone (Sigma-Aldrich, Bangalore, India) in concentrated H_2_SO_4_ as a reagent [23]. Spectrophotometric data were collected at 630 nm against a reagent blank. With 0–100 mg of glucose, a standard curve was plotted against various concentrations levels. The starch concentration was calculated by multiplying the resulting value by 0.9 for the conversion of glucose value to the starch [24]. The arsenomolybdate reagent (Sigma-Aldrich, Bangalore, India) was used to estimate reducing sugars using the alkaline copper technique, and absorbance was measured at 510 nm. A standard curve was established against graded pure glucose (0–50 mg) to assess sugar content reduction.

#### 2.7.3. Evaluation of Total Free Amino Acids, Proline, and Total Phenols

The total free amino acids were extracted from callus samples with 70% ethanol (Himedia, Mumbai, India), and calculated using ninhydrin reagent (Sigma-Aldrich, Bangalore, India) as described in [25]. A standard curve against glycine was used to compute the sum of total free amino acids (0–100 mg). The free proline content was extracted from callus using 3% sulphosalicylic acid using L-proline (Sigma-Aldrich, Bangalore, India)) as a reference solution and estimated according to [26]. Total phenols were evaluated by using 10% Folin phenol reagent and standard NaHCO_3_ (Himedia, Mumbai, India) according to [27]. The absorbance was measured using catechol as the standard in a spectrophotometer at 660 nm.

#### 2.7.4. Antioxidant Enzyme Assay

In vitro, one gram of fresh callus from the control, NaCl stress, and EMS treatments were homogenised in 2 mL of 50 mM potassium phosphate buffer (pH 7.5) (Himedia, Mumbai, India) containing 2 mM EDTA and were centrifuged at 15,000g for 20 min at 4 °C. Ammonium sulfate precipitation was used to concentrate the protein content in the supernatant, which was then filtered through Whatman No.1 filter sheets. A spectrophotometer (Mettler-Toledo, Mumbai, India) was used to determine the activity of chosen enzymes. The protein concentration at each fraction was determined by the method [28].

#### 2.7.5. Activity of Catalase

Catalase (EC 1.11.1.6) activity was determined using a modified version of the method described in [29]. The test was performed with 2.6 mL of 50 mM potassium phosphate buffer (pH 7.0), 0.4 mL of 15 mM H_2_O_2,_ (Himedia, Mumbai, India) and 0.04 mL of enzyme extract. The decrease in H_2_O_2_ absorbance at 240 nm coincided with the breakdown of H_2_O_2_. The enzyme activity was measured in units mg^−1^ protein (U = 1 mM of H_2_O_2_ reduction min^−1^ mg^−1^ protein).

#### 2.7.6. Activity of Peroxidase

The amount of soluble peroxidase (EC 1.11.1.7) activity was assayed by the method described by [30]. Two mL of 0.1 M phosphate buffer (pH 6.8), 1 mL of 0.01 M pyrogallol, Sigma-Aldrich, Bangalore, India) 1 mL of 0.005 M H_2_O_2_, and 0.5 mL of enzyme extract made up the POX assay combination. The amount of solution was incubated for 5 min at 25 °C, after which the reaction was stopped by adding 1 mL of 2.5 N H_2_SO_4_. The total amount of purpurogallin produced was calculated by measuring the absorbance at 420 nm against a blank prepared by adding the extract after the addition of 2.5 N H_2_SO_4_ at zero time. The activity was measured in the unit’s mg^−1^ protein. One unit (U) is defined as the change in the absorbance by 0.1 min^−1^ mg^−1^ protein.

#### 2.7.7. Superoxide Dismutase Activity

The amount of *superoxide dismutase* (EC 1.15.1.1) activity was determined according to method described by [31]. In 3 mLof (0.05 M) sodium phosphate buffer (pH 7.8), the reaction mixture of 1.17 × 10^−6^ M riboflavin, 0.1 M methionine, 2 × 10^−5^ M KCN, and 5.6 × 10^−5^ M Nitroblue tetrazolium salt (NBT) (Sigma-Aldrich, Bangalore, India) were dissolved. A total of 3 mL of reaction medium was added to 1 mL of enzyme extract. The reaction mixtures were illuminated in glass test tubes by two sets of Philips direct 40 W fluorescent lights in a single row. Elucidation was started at 30 °C for 1 h to initiate the reaction. Blank samples were kept in the dark for 30 min at the same time. The reduction was calculated based on changes in NBT. In the spectrophotometer, absorbance was calculated at 560 nm against the blank. One enzyme unit was defined as the volume of the enzyme extract corresponding to 50% inhibition of the process. The superoxide dismutase (SOD) activity was measured in units of mg^−1^ protein min^−1^.

#### 2.7.8. Statistical Analysis

In each subsequent test, the data values are given as the mean ± standard deviation (SD) for six plants. Duncan’s Multiple Range Test (DMRT) with a 0.05 percent significant criterion was to perform post hoc testing for intergroup comparisons. To assess the results, the SPSS software package for Windows (version 11.5; SPSS Inc., Chicago, IL, USA) was used to perform a one-way analysis of variance (ANOVA).

## 3. Results

### 3.1. NaCl’s Effect on Callus Growth and Plant Regeneration

Different concentrations of NaCl (0, 100, 200, 300, 400, and 500 mM) were used for assessing NaCl tolerance in callus formation in *A. vulgaris*. Calluses could tolerate maximum salinity (NaCl) stress up to 400 mM and could sustain for more than 30 days (Figure 1a–e), whereas for salinity at the 500 mM concentration, calluses began browning, and the death of the tissues takes place after a week (Figure 1f). Thus, all the experiments were conducted with calluses treated with NaCl concentrations between 0 and 400 mM. After 3 weeks of incubation, explants cultured on MS medium regenerated the callus and regeneration of the whole plant of *A. vulgaris* from control callus (0 mM NaCl stress) (Figure 2a,b).

### 3.2. Ethyl Methane Sulphonate (EMS) Treatment on Callus

To assess the high salinity (NaCl) stress, almost all callus growth was reduced after being treated with 0.5% of EMS for 30 min and 90 min was inhibited. In our study, it was noticed that 30 min exposure of 0.5% EMS treatment may be too short for causing mutagenesis in improving salt tolerance. However, the treatment of EMS for 90 min was found to be considerably more hazardous and had a detrimental influence on callus proliferation, whereas EMS treatment for 60 min resulted in a good survival rate of the callus. It suggests that the treatment of 0.5% EMS for 60 min is optimum for the induced mutation in the callus (Figure 3).

### 3.3. Shoot Elongation and Rooting

After 4 weeks of *A. vulgaris* callus treatment with 400 mM NaCl, it is further exposed to EMS treatment under in vitro stress conditions. Further, control and EMS-treated 400 mM NaCl calluses of *A. vulgaris* were grown on MS media containing 4.44 µM of N^6^-benzyladenine (BA) and 2.78 µM of TDZ for shoot elongation (Figure 2a and Figure 3a). Within two weeks of culture, shoot initiation was detected (Figure 2b,c and Figure 3b,c). Further, the initiated shoots were successfully elongated in 1.44 µM GA_3_ containing MS medium (Figure 2d and Figure 3d). The well-elongated shoots from the control and EMS-treated calluses were further cultured in MS medium supplemented with IAA after 4 weeks of inoculation to produce roots. Among different concentrations of IAA, 5.70 µM was found most successful for rooting (Figure 2e and Figure 3e). Successfully rooted plantlets were transferred to standard laboratory conditions (Figure 2f and Figure 3f).

### 3.4. Biochemical Analysis of Callus

#### 3.4.1. Effect of Induced Mutation by NaCl and EMS on Photosynthetic Pigments

The concentrations of chlorophyll and carotenoids in the callus of *A. vulgaris* were carried out, and total chlorophyll was expressed on unit fresh weight basis reduced by 25% following 400 mm NaCl treatment, as compared to the control. On an FW basis, a similar trend in carotenoids content was observed. At high salt concentrations (100–400 mm), chlorophyll and carotenoids levels were reduced in (Table 1). The pigment contents were increased when stressed calluses were treated with 0.5% EMS (Table 1).

#### 3.4.2. Effects on Carbohydrates (Total Sugar and Starch)

Carbohydrate contents in the calluses were measured, and at higher concentrations of NaCl, total sugar content increased gradually, and starch, on the other hand, decreased (Figure 4). Further, with mutagenic treatment with EMS, the contents of sugar and sucrose increased significantly (Figure 4). Moreover, the total sugar content was high at 400 mM NaCl, whereas the starch concentration was observed maximum at 100 mM NaCl under EMS treatment.

#### 3.4.3. Effects on the Free Amino Acid, Proline, and Total Phenol

Salinity (NaCl) stress decreases in the free amino acid pool, whereas the endogenous proline content increased under NaCl stress, with the maximum increase observed at 300 mM. The maximum increase in free amino acids was observed in stressed calluses treated with EMS (mutagenic agent) (Table 2). Under saline conditions, EMS-treated calluses showed a significant increase in total phenol over control, and NaCl stressed callus (Table 2).

#### 3.4.4. Effects on Antioxidant Enzyme Activities

In the callus of *A. vulgaris*, the effect of mutations induced by NaCl and EMS on the total activity of three antioxidant enzymes, including Catalase (CAT), Peroxidase (POD), and Superoxide dismutase (SOD), was investigated and examined in the callus of *A. vulgaris*. As expected, the results showed a steady increase in the overall activity of POD and SOD from 100–400 mM NaCl, whereas CAT activity decreased (Figure 5). We recorded a further significant increase in CAT, POD, and SOD with mutagenic effects of EMS treatment. Furthermore, although POD and SOD activity peaked at 200 mM NaCl, CAT activity was higher in 100 mM NaCl (Figure 5).

## 4. Discussion

One of the key environmental elements that affect the plants is salinity. Salt concentrations in the soil have a strong negative influence on all crop species growth and output. However, as many studies have shown, a crop’s ability to endure and thrive in salt conditions varies widely among different species and varieties. In our present investigation, calluses developed from nodal stem explants were used with five different concentrations for assessing salt (NaCl) tolerance in *A. vulgaris.* Salinity stress caused a significant effect on the growth and development of callus [32]. In our present study, NaCl-stressed calluses were successfully regenerated with 0.5% EMS for 60 min, exhibiting the most robust salt tolerance (Figure 2). It is possible that 60 min of exposure was the best length for 0.5% EMS treatment, which could stimulate the maximum frequency of mutagenesis for salt tolerance, were reported by [33]. However, salt tolerance in sugarcane mutants were developed from calluses mutated with EMS were previously observed [34].

In *A. vulgaris* chlorophyll and carotenoid levels were lowered by salinity but slightly increased with the mutagenic effects of NaCl and EMS combinations, a progressive decrease in chlorophyll a, chlorophyll b, total chlorophyll, and carotenoid was observed with NaCl alone (Table 1). The reduction in chlorophyll concentration under NaCl stress due to changes in the lipid-protein ratio of the pigment-protein complex activity was previously reported by [35]. Similar results were previous reported on decreased contents of chlorophyll and carotenoids by (NaCl) salinity stress [36,37]. However, the EMS-treated callus showed a further increase in photosynthetic pigments, which directly implies the effectiveness of EMS on pigments of *A. vulgaris*.

In *A. vulgaris,* total sugar content increased by combining the mutagenic effects of salinity and EMS than NaCl alone (Figure 4). Our results are supported by previous reports that salt stress increases, glucose, sucrose, and fructans in many plants [38]. The activity of sucrose phosphate synthase increases as the concentration of reducing and non-reducing sugars rises in salinity, whereas starch phosphorylase activity decreases [39]. The mutagenic effects of EMS on NaCl-stressed calluses showed a further increase in contents of carbohydrates. We found a further increase in total amino acid when NaCl stressed the callus of *A. vulgaris* treated with EMS. Increased content of proline and total phenol was observed under salinity stress in *A. vulgaris* (Table 2).

Plants under stress, meanwhile, develop defensive systems to counteract the harmful consequences of oxidative stress. One of the most common defense responses to abiotic stressors is ROS scavenging [40]. The primary enzymatic ROS scavenging activities include catalase (CAT), superoxide dismutase (SOD), and peroxidase (POD). SOD may catalyse the dismutation of superoxide radicals to generate O_2_ and H_2_O_2_, the produced H_2_O_2_ eliminated by POD and CAT in *Pisum sativum* and *Brassica chinensis* reported by [41].

The results of this investigation revealed that all three antioxidant enzymes were active in the callus of *A. vulgaris*. Under salinity (NaCl) stress situation, anti-oxidative stress ability was linked. The anti-oxidative response varies based on the species, the plant’s development and metabolic condition, and the length and intensity of the stress. The activity of SOD and POD increased under NaCl stress. Similarly, catalase activity was found lower than SOD and POD (Figure 5). Interestingly, the high SOD activity coincided with changes in the specific activities of POD assay. However, catalase activities were suppressed under salinity (NaCl) stress. Additionally, the combination of mutagenic effect of EMS and NaCl treatment showed a significant rise in the activities of CAT, POD, and SOD over control and NaCl stress (Figure 5).

## 5. Conclusions

According to the findings of this study, it can be stated that salinity had an impact on *A. vulgaris* growth and regeneration. Treatment with EMS had a mutagenesis impact, which reduced the toxic effects of NaCl stress and resulted in increased growth and biochemical activities. The improved salt tolerance of mutants implies that *A. vulgaris* mutagenesis with EMS is significant. This approach could be effective for mutation breeding and the creation of salt-tolerant medicinal plants for successful cultivation, thereby providing raw material to pharmaceutical companies and local medicinal purposes.

## Figures and Tables

**Figure 1 biotech-10-00018-f001:**
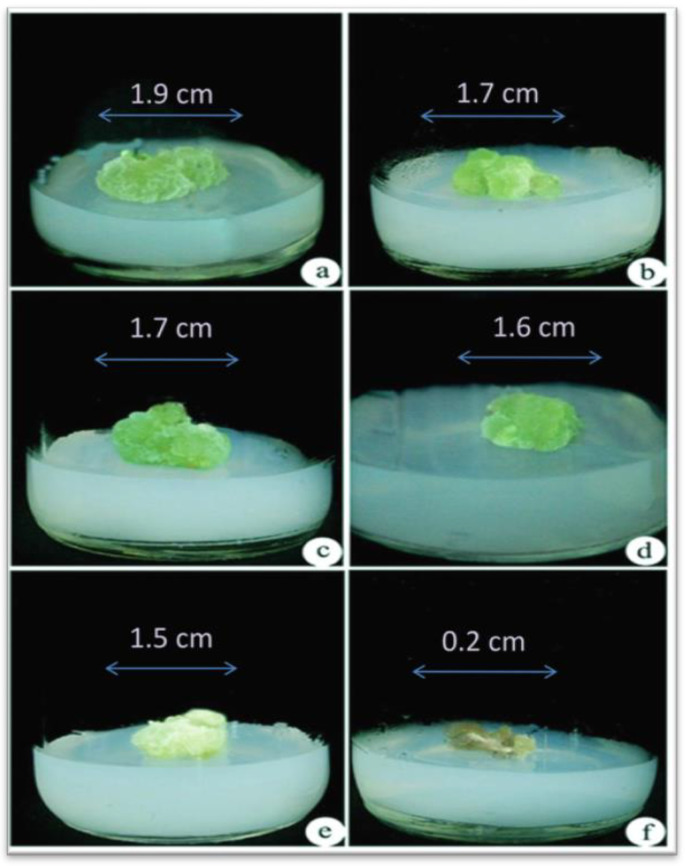
Assessment of NaCl tolerance in callus of *A. vulgaris*. (**a**) Control callus culture from (0 mM NaCl), (**b**)callus cultured on (100 mM NaCl), (**c**) callus cultured on (200 mM NaCl), (**d**) callus cultured on (300 mM NaCl), (**e**) callus cultured on (400 mM NaCl) showed its growth reduction and discoloration, (**f**) dead calluses at (500 mM NaCl) concentration after 30 days.

**Figure 2 biotech-10-00018-f002:**
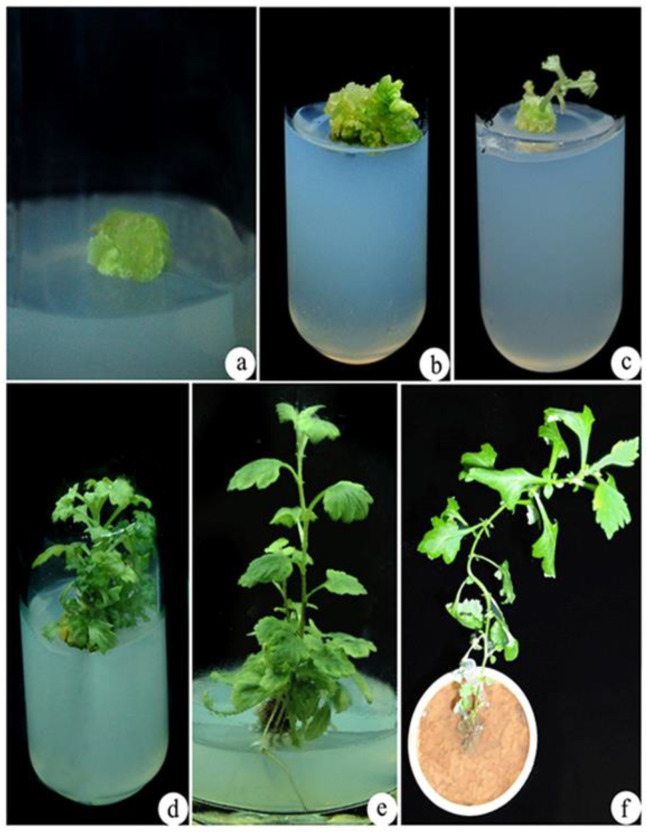
In vitro regeneration of *A. vulgaris* from control callus (0 mM NaCl stress). (**a**) Regenerated callus from nodal explants, (**b**,**c**) shoot induction from the callus, (**d**)multiple shoots proliferation, (**e**) elongated shoot with roots, (**f**) hardened plant in the cup.

**Figure 3 biotech-10-00018-f003:**
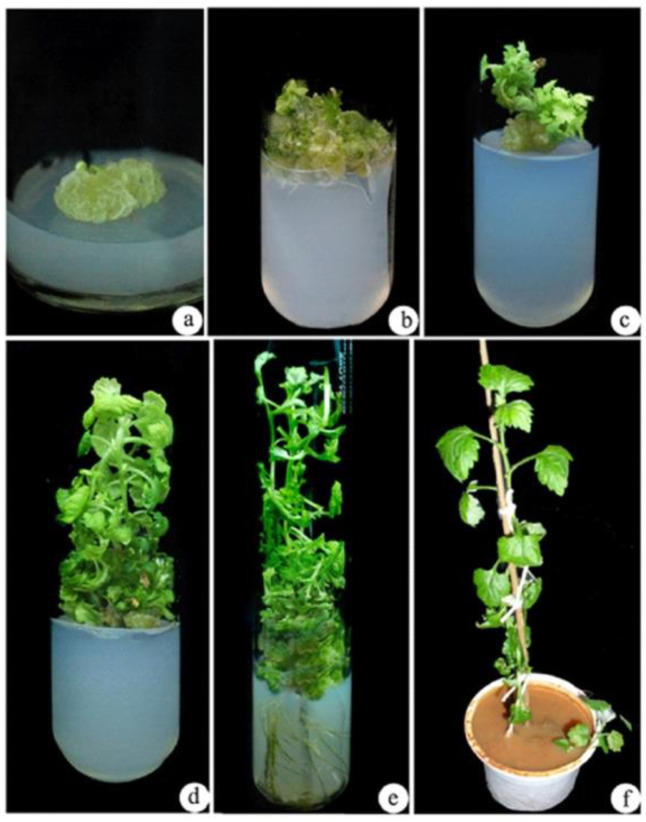
Influence of ethyl methane sulphonate (EMS) 0.5% on the growth of mutants from callus of *A. vulgaris* on NaCl stressed media. (**a**) Regenerated callus, (**b**,**c**) shoot induction from callus, (**d**) multiple shoots induction and elongation from callus, (**e**) multiple shoot induction with roots, (**f**) hardened plant in a cup.

**Figure 4 biotech-10-00018-f004:**
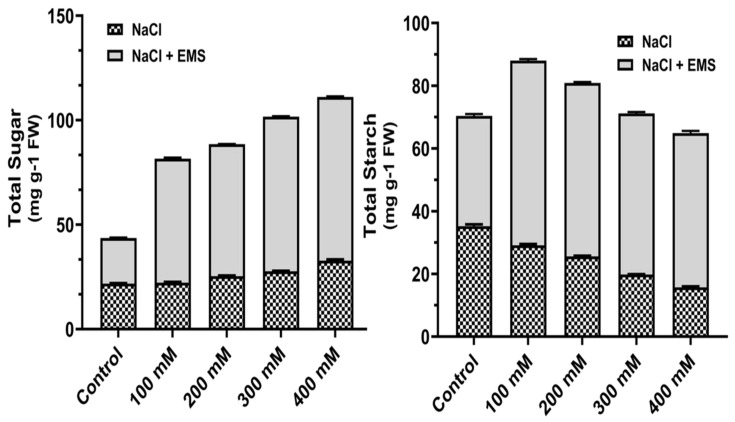
Mutagenic effect of sodium chloride (NaCl) and ethyl methane sulphonate (EMS) on carbohydrates (total sugar and starch) contents of callus of *A. vulgaris*. The values reflect the mean standard deviation of three independent experiments. According to DMRT, means separated by different letters are significantly different from each other at 5% level of significance (*p* ≤ 0.05).

**Figure 5 biotech-10-00018-f005:**
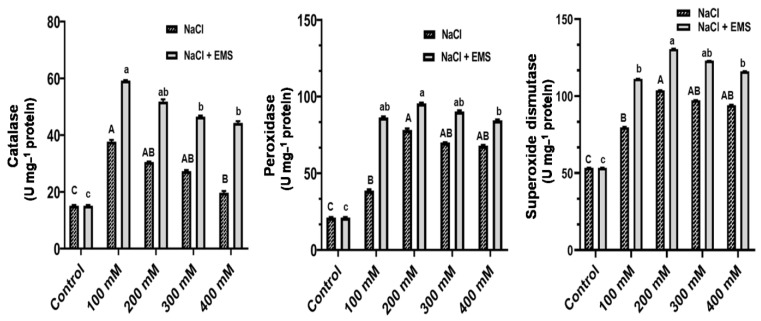
Antioxidant enzymes activities of (CAT, POD, and SOD) on NaCl and ethyl methane sulphonate induced mutation treated on calluses of *A. vulgaris.* The values reflect the mean ± SE of three repeated experiments. Different letters correspond to a significant difference from each other at (*p* ≤ 0.05), according to DMRT. Uppercase for NaCl and lowercase for (NaCl + EMS).

**Table 1 biotech-10-00018-t001:** Mutagenic effect of sodium chloride (NaCl) and ethyl methane sulphonate (EMS) on photosynthetic pigments of callus of *A. vulgaris.* The values reflect the mean standard deviation of three independent experiments. According to DMRT, means separated by different letters are significantly different from each other at 5% level of significance (*p* ≤ 0.05).

Treatment	Chlorophyll a (mg g^−1^ FW)	Chlorophyll b (mg g^−1^ FW)	Total Chlorophyll (mg g^−1^ FW)	Carotenoid (mg g^−1^ FW)
**Control**				
0 mM	0.70 ± 0.04 c	0.34 ± 0.07 e	1.04 ± 0.04 b	0.58 ± 0.008 cd
**NaCl Stress**				
100 mM	0.67 ± 0.06 c	0.30 ± 0.04 ef	0.97 ± 0.02 cd	0.54 ± 0.003 d
200 mM	0.61 ± 0.03 cd	0.28 ± 0.01 ef	0.89 ± 0.01 d	0.49 ± 0.001 de
300 mM	0.57 ± 0.04 cd	0.24 ± 0.02 f	0.81 ± 0.07 de	0.48 ± 0.009 de
400 mM	0.54 ± 0.02 d	0.20 ± 0.01 f	0.74 ± 0.03 e	0.41 ± 0.004 e
**NaCl + EMS (0.5%)**				
100 mM	0.97 ± 0.08 b	0.57 ± 0.01 cd	1.54 ± 0.06 a	0.79 ± 0.004 bc
200 mM	0.93 ± 0.05 b	0.51 ± 0.07 d	1.44 ± 0.06 a	0.79 ± 0.007 bc
300 mM	0.88 ± 0.07 bc	0.51 ± 0.04 d	1.39 ± 0.02 ab	0.73 ± 0.003 bc
400 mM	0.84 ± 0.03 bc	0.46 ± 0.05 de	1.30 ± 0.01 ab	0.69 ± 0.008 c

**Table 2 biotech-10-00018-t002:** Mutagenic effect of sodium chloride (NaCl) and ethyl methane sulphonate (EMS) on total free amino acids, proline, and total phenol contents of callus of *A. vulgaris.* The values reflect the mean standard deviation of three independent experiments. According to DMRT, means separated by different letters are significantly different from each other at 5% level of significance (*p* ≤ 0.05).

Treatment	Total Amino Acid (mg g^−1^ FW)	Proline (mg g^−1^ FW)	Total Phenol (mg g^−1^ FW)
**Control**			
0 mM	29.8 ± 0.3 c	43.0 ± 0.2 cd	27.3 ± 0.3 c
**NaCl Stress**			
100 mM	25.8 ± 0.6 c	54.7± 0.8 b	25.6 ± 0.6 c
200 mM	20.6 ± 0.2 c	59.2 ± 0.9 b	21.7 ± 0.3 c
300 mM	15.8 ± 0.4 cd	69.4 ± 0.1 a	19.4 ± 0.2 cd
400 mM	10.7 ± 0.6 d	63.3 ± 0.4 ab	14.1 ± 0.4 d
**NaCl + EMS (0.5%)**			
100 mM	48.3 ± 0.2 a	45.2 ± 0.6 c	51.6 ± 0.2 a
200 mM	43.2 ± 0.8 ab	48.6 ± 0.4 bc	45.5 ± 0.1 ab
300 mM	39.2 ± 0.4 b	51.3 ± 0.7 bc	40.0 ± 0.1 b
400 mM	35.0 ± 0.6 bc	44.4 ± 0.5 c	33.4 ± 0.3 bc

## Data Availability

Not applicable.

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
