# Peer review of "Impact of Ethyl Methane Sulphonate Mutagenesis in Artemisia vulgaris L. under NaCl Stress"

_biotech, 2021, doi:10.3390/biotech10030018_

Round 1

Reviewer 1 Report

The manuscript is reporting effects of in vitro EMS mutagenesis to increase tolerance against salt stress in Artemisis and analyses metabolic changes in regenerated mutagenized plants. This analysis isbased on the assessment of some biochemical parameters, including the content of chlorophylls, sugars and starch and the activities of enzymes involved in mitigation of oxidative stress.

The results reported in the paper are certainly interesting, however the manuscript needs to be thoroughly revised for clarity and language before it can be regarded to be fit to print. I would suggest that the authors take another turn at the manuscript to revise all parts to enhance intelligibility of the text and that a thorough language editing is provided to increase the readability and to correct the spelling.

I include a number of detailed observations and comments, hoping that these are valuable for the required revision.

Detailed comments

Abstract

Pls revise entirely to provide an improved account of the work. Some parts read like the result section and not like a concise summary of the manuscript. Pls. also revise language and wording, as the text in its current form is hard to understand, e.g. L10-11, L16-17.
Also introduce abbreviations (POD, SOD CAT) or mention them in keywords. Also introduce new abbreviations at first mention in rest of manuscript.

Keywords

Pls. revise for relevance and significance – e.g. why is NBT included? Others seem missing Delete source or rephrase

Introduction

Pls. revise first paragraph L28-35 regarding grammar and intelligibility.

L44 correct to: compartimentalization

Pls. further revise: L47-58: e.g. capable, sustain (?) Ox stress, production .. is produced …

L56: revise: suggestion: The most important enzymatic ….

L61: delete “previous reports show that”

L65: synchronized??? Explain or revise!

L66: delete first comma

L67: with limited space and time??? Explain or revise!

L71: revise start of sentence

L74: correct: traits not trailsL75-76 revise last part of sentence

Materials/Methods

L87: Surface sterilisation of seeds is first step, revise order

L90: Delete “source” or rephrase

L97: correct to “autoclaved”

Pls. correct spelling and use of capital letter throughout the manuscript: e.g. L102, L315, L326, …

L106-107: Rephrase!

L116: concentration of what? Is singular (callus) correct?

L120: supported??? Rephrase

L127: suggest: … the adventitious roots were washed …

L.143: include full stop after and start a new sentence: The alkaline….

L148-149: Revise!

L149 and following (152, 153, 162, 170, 1080, 309, …): The indication of referenced work with numbers results in awkward reading: e.g. following (25), method of (26), …
Either revise or indicate the authors followed by the number of reference.

L158ff: The authors compare control, salt stressed and EMS treated plants. However the EMS plants were also initiated under high NaCl and therefore salt stressed. Explain the approach and better indicate the compared treatments.

L182 -186: revise and rephrase for grammar, clarity!

L193: revise to: Subsequent testing ….

Results

L204ff: Revise language- grammar / wording, use plural (calli)

L207: revise: experiments were conducted with calli treated with NaCl concentrations between ….

Fig1 caption: indicate time of growth on NaCl medium, revise wording (“showed”)

L222-228: Revise for clarity, grammar and readability!

L248ff: Pls. revise text to clarify that the reported effects were not directly due to EMS treatment but resulted from the mutations induced by previous EMS exposure! This is relevant for text as well as table captions!

L263: revise

L270: Revise sentence – While???, found???

L271: Revise to: The maximum increase  …

L279, 284: see above: the effects of the mutations induced by EMS, not the EMS treatment itself!

Discussion

L293: delete harvestable (seems to be a pleonasmus)

L291-292: indicate when and where plants are exposed

L293-295: Revise for language and clarity – seems to be rather results than discussion.

L295-303: Pls revise thoroughly!

L302: rephrase, “similarly” seems to be wrong

L305/314/318/337: see above cf. L248 – see wording in L341

L307f: revise for readability ( …might be…), especially the second part of the sentence!

L330: Revise to: The present study shows that ….

L330ff: Revise for intelligibility!

L334: “Whereas” seems to be wrong –revise

L337-338: Revise for intelligibility

Conclusions

L342: “utilizing” is wrong (suggest: … and led to….)

Reviewer 2 Report

biotech-1294860: Impact of Ethyl Methane Sulphonate (EMS) Mutagenesis in Artemisia vulgaris L. under NaCl Stress

The article deals with the mutational influence of ethyl methane sulphonate (in vitro mutagenesis, chemical mutagen) and subseguent plant regeneration in the plant species Artemisia vulgaris L. The resulting mutants were found to have a better tolerance to the effects of stress from sodium chloride (tested range 0-400 mM NaCl). The whole complex system of physiological parameters was monitored and the results were demonstrable. It was found, that the use of ethyl methane sulphonate may be useful for mutation breeding and develop ment of salt-tolerant medicinal plants for successful cultivation. 
The article brings new scientific messages, is goot compiled, and the reader is well versed in the manuscript. The article is burdened with many minor typos and errors, which work very badly in the context of editing. I have just some small recommendations for improving the manuscript. 

Comments:

  • Title: „Impact of Ethyl Methane Sulphonate (EMS) Mutagenesis in Artemisia vulgaris L. under NaCl Stress“ is supposed to be „Impact of Ethyl Methane Sulphonate Mutagenesis in Artemisia vulgaris L. under NaCl Stress“
  • Line 11: „ethyl methane sulphonate“ is supposed to be „ethyl methane sulphonate (EMS)“
  • Line 36: „artemisia annua L“ is supposed to be „Artemisia annua
  • labeling the plant species with its complex scientific name (Artemisia vulgaris ) throughout the manuscript is unnecessary, the name may be: A. vulgaris
  • line 198: „1. Effect of NaCl on callus growth and Plant regeneration“ is supposed to be „3.1. Effect of NaCl on callus growth and plant regeneration“
  • line 202: „at 25±1Ëšc.“ is supposed to be „at 25±1 ËšC.“
  • Table 1 + Table 2: „Nacl Stress“ is supposed to be „NaCl Stress“

Reviewer 3 Report

This paper demonstrates the change in the biochemical parameters of mugwort Artemisia vulgaris L. callus cultured on salt and combination of salt and a mutagenic agent ethyl methane sulphonate (EMS). The results obtained in the manuscript are interesting for the scientific community and can be used to create salt-resistant medical plants. Therefore, the topic of this manuscript is relevant for BioTech. However, before this manuscript can be published, some improvements should be performed.

General remarks:

  • All the above biochemical analyses were carried out on callus vulgaris. Perhaps the authors should conduct similar analyses on plants obtained from mugwort callus? Did the biochemical parameters differ in the regenerating plants?

  • Did authors analyze the biochemical parameters of callus under the influence of EMS without salt?

  • In the abstract, the authors write “Thus the mutants were more salt-tolerant than control plants”. Have the authors conducted experiments on the resistance to salt stresses of mugwort plants obtained from calluses? Did the authors compare the resistance of plants to salt stress of plants obtained from common mugwort callus, regenerating from callus under the influence of plant salt, and plants from callus after EMS treatment and growing on a medium with salt?

  • In the chapter 2. Ethyl methane sulphonate (EMS) treatment on callus, the authors write «The result shows that 30 mins of 0.5% of EMS treatment might be too short of causing mutagenesis in improving salt tolerance… But, EMS treatment for 60 mins resulted in good survival of callus». It is not entirely clear to me how the authors assessed the level of mutagenesis? How was the mutation rate estimated? Why was 30 minutes not enough?

  • In the conclusion, the authors write «Hence, from the results of the present study, it can be concluded that salinity affected the growth and regeneration of Artemisia vulgaris ». It is not clear how salt stress affected the growth and regeneration of plants? Probably, the authors should supplement the results of the manuscript with a description of plants obtained from callus. For example, the weight and height of plants, the size of leaves, the length of roots, etc. Compare the characteristics of the obtained plants with the control plants.

  • Why did the authors analyze only the biochemical parameters of callus cultured on a MS medium with 400 mÐœ of salt? Why didn't analyze the parameters of the callus that grew on a medium with 100, 200 and 300 mM of salt? Accordingly, did not get plants from these calluses?

The main topic of the criticism is that this Ms contains numerous misprints, mistakes in English grammar and in the writing style. I recommend that the authors should use some help of a native English speaker or send the manuscript to an English Editing Service that proofreads scientific writing.

It is very difficult to understand different parts of the manuscript in its present form.

Minor remarks:

On page 1: line 1, add an abbreviation ethyl methane sulphonate (EMS);

                   line 35, correct “artemisia annua L” to “Artemisia annua L.”;

On page 2: line 76, perhaps authors should rewrite the sentence, the phrase is duplicated in vitro in a row. «…which can be easily controlled in vitro. In vitro tissue culture constitutes an important tool…»;

                line 80, give the full name of the substance and its abbreviation in parentheses,

 ethyl methane sulphonate (EMS). And further in the text, use only the abbreviation of this substance. I think authors should check the text on the same mistakes;

                line 86, correct “Artemisia vulgaris L.” to “A. vulgaris”, because Artemisia used before, check the text on similar mistakes,

                line 93, references should be given at the end of the sentence. Probably the authors should introduce the abbreviation of the nutrient medium Murashige and Skoog (MS);

On page 3: line 113, check the composition of the nutrient medium. Perhaps it was mentioned earlier. Do not write the full name of phytohormones if the abbreviations of phytohormones were given in parentheses earlier in the text;

               line 116, correct “400 mM concentration” to “400 mM NaCl”;

               line 120, reduce the composition of the environment, as it is described earlier;

               line 120, write the full name of the phytohormone TDZ;

               line 122, write the full name of the phytohormone GA3;

On page 5: line 199-202, these suggestions should be moved to the materials and methods chapter;

               line 208-209, remove repetition of the description of the nutrient medium;

Fig.1,2,3 add a dimension scale, line and top how many cm;

On page 7: line 237, “…observed within 2 weeks (Figure 1b-c & 3b-c)...” perhaps the authors meant (Figure 2b-c & 3b-c);

Sections 3.3 and 3.4 should be moved to the Materials and methods chapter.

Table 1 and Table 2, correct “Nacl Stress” to “NaCl”;

In the list of references, authors should make a single line spacing.

After removal of all the above comments, the article can be published in BioTech.

Round 2

Reviewer 1 Report

Thanks to the authors for addressing my comments and for the response provided on their revision.

Unfortunately a lot of editing errors still remain or have been introduced with the revisions. Therefore I strongly recommend that the language in the abstract as well as in the other parts of the manuscript is checked again by an English native speaker, e.g. via the respective services offered by MDPI. The text needs to be checked for language, grammar, typos, missing fullstops, wrong capitals and many other editorial errors.

While this is not comprehensive I point out a few errors as examples:

Remove superfluous blanks (e.g. L35, L51, etc.)

Correct typos (e.g. L13, L25, etc.)

Insert missing fullstops (e.g. L65, etc.)

Check article use – articles are missing in (e.g. L23, etc.)

Use abbreviations in text after they have been introduced initially and introduce abbreviations consistently (e.g. L38)

Check and correct use of capital letters (e.g. L23, L131, L298, etc.)

Correct grammar, L11-12: its “ to develop” not “to developed”, also e.g. L32, L74, L95, L240, L298, etc.

Correct spelling (e.g. L125, etc.)

Check with the editorial office if the style of referencing is correct: e.g. “by anthrone–sulphuric acid method of [23]”
I would recommend using the following way of citation in all of these cases: “by the anthrone–sulphuric acid method developed/described by Chow and colleagues [23]” I.e. to include the referenced authors followed by the numerical reference in such cases rather than use the numericals alone. Pls. check back with the editorial office and revise accordingly!

Pending editorial corrections supervised by the editorial office I am fine with publication.

Author Response

Dear Reviewer,

                     Thanks for the comments and suggestions for the manuscript. As per the comments for the minor review report, I have corrected and responses that were attached in the word file.

Thanks for the cooperation and comments.

With Regards,

Reviewer 2 Report

Thank you for accepting all the controversial points of the previous version of the manuscript.

Author Response

Reviewer-2 

Didn't comment on any suggestions on the manuscript, Reviewer-2 has accepted my previous version of the manuscript.

Thanking you,